# High-Throughput Sequencing Reveals New Viroid Species in *Opuntia* in Mexico

**DOI:** 10.3390/v16081177

**Published:** 2024-07-23

**Authors:** Candelario Ortega-Acosta, Daniel L. Ochoa-Martínez, Esteban Rodríguez-Leyva

**Affiliations:** 1Posgrado en Fitosanidad-Fitopatología, Colegio de Postgraduados, Texcoco C.P. 56264, Estado de México, Mexico; ortega.candelario@colpos.mx; 2Posgrado en Fitosanidad-Entomología y Acarología, Colegio de Postgraduados, Texcoco C.P. 56264, Estado de México, Mexico; esteban@colpos.mx

**Keywords:** viruses in *Opuntia*, cactaceae, new viroids, Mexico, *Pospiviroidae*, diversity

## Abstract

In the main cactus pear (*Opuntia ficus-indica*)-producing region in the State of Mexico, fruit production occupies the largest cultivated area with 15,800 ha, while 900 ha are cultivated for edible young *Opuntia* pads (“nopalitos”) which are consumed as vegetables. Two composite samples consisting of cladodes of plants for fruit production (*n* = 6) and another of “nopalitos” (*n* = 6) showing virus-like symptoms were collected. Both sample sets were subjected to high-throughput sequencing (HTS) to identify the viruses and viroids. The HTS results were verified using RT-PCR and Sanger sequencing. Subsequently, 86 samples including cladodes from “nopalitos”, plants for fruit production, xoconostles, and some wild *Opuntia* were analyzed via RT-PCR with specific primers for the viruses and viroids previously detected via HTS. Three viruses were discovered [Opuntia virus 2 (OV2), cactus carlavirus 1 (CCV-1), and Opuntia potexvirus A (OPV-A)], along with a previously reported viroid [Opuntia viroid 1 (OVd-1)]. Additionally, two new viroids were identified, provisionally named the Mexican opuntia viroid (MOVd, genus *Pospiviroid*) and Opuntia viroid 2 (OVd-2, genus *Apscaviroid*). A phylogenetic analysis, pairwise identity comparison, and conserved structural elements analysis confirmed the classification of these two viroids as new species within the *Pospiviroidae* family. This is the first report of a pospiviroid and two apscaviroids infecting cactus pears in the world. Overall, this study enhances our understanding of the virome associated with cactus pears in Mexico.

## 1. Introduction

*Opuntia ficus-indica* (L.) Miller or the cactus pear is native to Mexico [1] and is currently the most widely cultivated cactus in the world [2]. Globally, it is primarily cultivated as forage for livestock feed, for fruit production, and to prevent erosion in desert regions [2,3]. In Mexico, in addition to its fruit, the immature cladodes are consumed as vegetables known as “nopalitos” [2,4]. This species also has applications in the food industry and serves as a host for *Dactylopius coccus*, which is used in the production of carminic acid, a dye employed in food, textile, and pharmaceutical products [2,5].

The State of Mexico is the leading producer of cactus pears in Mexico, with a cultivated area of approximately 17,500 ha. Of these, 15,800 ha are dedicated to cactus pears for fruit production (tuna), 900 ha are dedicated to the production of nopal as a vegetable (=nopalitos), and 800 ha are dedicated to the production of xoconostles (*Opuntia joconostle*), a plant that produces a specific kind of acidic fruit for domestic consumption [6]. The greatest production of cactus pears is concentrated in the eastern region of the State of Mexico, particularly in the municipalities of San Martín de las Pirámides, Axapusco, Otumba, Nopaltepec, Teotihuacan, Temascalapa, and Acolman [6].

High-throughput sequencing (HTS) or next-generation sequencing has significantly increased virus detection [7] and revolutionized the study of nucleic acids by allowing the sequencing of millions of nucleotides in a short period of time with very high redundancy (sequencing depth) [8]. When combined with specific bioinformatics tools, HTS can be used for the detection of known viruses as well as the discovery of new viruses or viroids [8,9]. One of the major advantages over classical detection techniques such as the enzyme-linked immunosorbent assay (ELISA) and various PCR methods is that HTS does not require prior knowledge of viral genomic information, and it allows for identification of the plant’s entire virome in a single assay [10].

Despite Mexico being the center of origin and domestication of *O. ficus-indica*, as well as many other species of the genus *Opuntia* [1], there are limited studies regarding virus detection in these plants. In studies using high-throughput sequencing for virus detection in cactus pears conducted in State of Mexico and Mexico City [11,12], Opuntia virus 2 (OV2, *Tobamovirus* genus), cactus carlavirus 1 (CCV-1, *Carlavirus unicacti*), and Opuntia potexvirus A (OPV-A, *Potexvirus* genus) have been identified. Additionally, a novel viroid provisionally named Opuntia viroid 1 (OVd-1, *Apscaviroid* genus) was discovered, with the latter being found in nopalitos in Mexico City [12].

In a study conducted on various wild species of the genus *Opuntia* collected from United Mexican States to determine the presence of DNA genome viruses, the squash leaf curl virus (SLCV, *Begomovirus cucurbitapeponis*) and watermelon chlorotic stunt virus (WmCSV, *Begomovirus citrulli*) were found [13]. Additionally, a new geminivirus tentatively named Opuntia virus 1 (OpV1, *Opunvirus opuntiae*) was detected in *Opuntia santa-rita* in the state of Sonora, Mexico [14]. Other viruses detected in cactus pears in the State of Mexico through RT-PCR include Schlumbergera virus X (SVX, *Potexvirus ecschlumbergerae*) and rattail cactus necrosis-associated virus (RCNaV, *Tobamovirus muricaudae*) [15,16]. However, to date, no comprehensive study has been conducted to determine the viruses present in the State of Mexico, which is the main region where cactus pears, xoconostles, and to a lesser extent, nopalitos are produced in the country.

In this study, we utilized HTS to investigate the virome of both cactus pears for fruit production and nopalitos. Previously, employing this approach, a new virus and a new viroid species were detected in nopalitos [12]. In the present study, we leveraged the advantages of HTS technology to delve deeper and characterize the viruses and viroids in the primary production region of cactus pear production regions in the State of Mexico. Our results identified various viruses and a viroid previously reported in nopalitos. Additionally, we detected two new viroids, one provisionally named the Mexican opuntia viroid (MOVd, *Pospiviroid* genus) in nopalitos and another provisionally named Opuntia viroid 2 (OVd-2, *Apscaviroid* genus) in cactus pears for fruit production. This study sheds light on the diversity of viruses and viroids in *Opuntia* plants in Mexico.

## 2. Materials and Methods

### 2.1. High-Throughput Sequencing (HTS) and Bioinformatic Analysis

From June to December 2023, cladodes from cactus pears (*Opuntia ficus-indica*) and nopalitos with different putative virus-like symptoms were collected in Cuautlacingo, municipality of Otumba, and San Felipe Teotitlán, municipality of Nopaltepec, located in the eastern region of the State of Mexico. The main symptoms consisted of chlorosis with green islands in nopalitos (Figure 1A,B) and mottling and chlorotic spots in cactus pears (Figure 1D). Total RNA extraction from each sample was performed using the SV Total RNA Isolation System kit (Promega, USA) following the manufacturer’s instructions. The quality and quantity of RNA were verified using a NanoDrop spectrophotometer (Thermo Fisher Scientific, Waltham, MA, USA), and RNA integrity was assessed by agarose gel electrophoresis (1%) and staining with ethidium bromide. Approximately 1.5 µg of total RNA was extracted from each collected symptom to form a composite sample (*n* = 6) for nopalitos and a composite sample (*n* = 6) for cactus pears. The two composite samples were sent to Innomics Inc. (San Jose, CA, USA) where the DNBSEQ Eukaryotic Strand-specific mRNA library was prepared, and sequencing was performed on the DNBSEQ platform (150 bp PE).

The sequences from each sample were *de novo* assembled using SPAdes v3.15.2 software with default parameters [17]. The obtained contigs were then subjected to a search in a local virus database (reference genomes, year 2023) using BLASTn and BLASTx. Subsequently, the reduced list of viral hits was compared against the complete nonredundant (nr) GenBank databases using BLASTn and BLASTx. For genome reconstruction, bowtie2 [18], SAMtools [19], and the Integrative Genomic Viewer (IGV) version 2.3 [20] were utilized. The assembly characteristics of the obtained genomes were evaluated with Qualimap 2 [21] and coverage and read depth maps were visualized and obtained through Geneious version 2024.0 (Biomatters). Finally, the genomes were annotated with the assistance of ORFinder (NCBI) and deposited into GenBank.

### 2.2. Genomic Analysis of New Viroid Species

For the detected viroids, a multiple-sequence alignment was performed using ClustalW implemented in MEGAX [22], utilizing a representative sequence of opuntia viroid-2 (OVd-2, GenBank accession number: PP579661), the Mexican opuntia viroid (MOVd, GenBank accession number: PP579660), and all RefSeq sequences of the *Pospiviroidae* family available in GenBank. A phylogenetic tree was constructed with 1000 bootstrap replicates using the maximum likelihood method and the Jukes–Cantor model in MEGAX [9,23]. The phylogenetic tree was visualized and edited using Interactive Tree of Life (iTOL) [24].

To compare the results with the taxonomic demarcation criteria approved by the International Committee on Taxonomy of Viruses (ICTV), pairwise sequence comparisons were performed using the Sequence Demarcation Tool (SDT) version 1.3 [25]. The pairwise identity graphs produced by SDT and the color-coded distance matrices were compared with the appropriate demarcation criteria to determine whether the viroid sequences found in the present study fell within the limits of existing species or should be considered new species.

The minimum free energy secondary structures for the two viroids were predicted using the UNAFold [26] web server (available online: www.mfold.org, accessed on 5 January 2024). The obtained secondary structures were further edited for printing using RnaViz 2 [27].

### 2.3. Genomic Analysis of the Detected Viruses

For the phylogenetic analyses of the viruses detected in the viral metagenomic analysis, sequences from representative members of the genus to which they belong were retrieved from GenBank. In all cases, multiple-sequence alignments were performed using MAFFT [28], phylogenetic inference was conducted using the maximum likelihood method with IQ-tree 2 [29], and ModelFinder was used for evolutionary model selection [30]. Nodal support was estimated using ultrafast bootstrapping (UFBoot) (1000 replicates) [31], and phylogenetic trees were visualized and edited in iTOL [24].

### 2.4. Validation of the Detected Viruses and Viroids

To validate the viral metagenome of cactus pears, different primers (Table 1) were designed based on sequences obtained from HTS. For cDNA synthesis, 4 μL of the composite RNA sample analyzed by HTS, 0.5 μL of random primers (Promega, Madison, WI, USA), and 5.5 μL of nuclease-free water were used. The reaction mixture was incubated at 70 °C for 5 min and then cooled on ice for one min. Subsequently, 5 μL of M-MLV buffer (5X), 2.5 μL of dNTP mix (10 mM), 0.25 μL of M-MLV reverse transcriptase (200 U/μL; Promega, Madison, WI, USA), and 5 μL of nuclease-free water were added. The tubes were incubated at 37 °C for 60 min, followed by 72 °C for 10 min. The transcribed cDNA (1 µL) was used as a template for PCR in a mixture containing 2 μL of reaction buffer (5X Green GoTaq^®^), 0.6 μL of MgCl2 (25 mM), 0.2 μL of dNTPs mix (10 mM), 0.6 μL (10 μM) of each primer, 4.9 μL of nuclease-free water, and 0.1 μL of GoTaq^®^ DNA Polymerase (5 U/μL; Promega, Madison, WI, USA). The PCR conditions consisted of an initial incubation at 95 °C for 3 min, followed by 35 cycles of denaturation at 95 °C for 30 s, annealing at 55 °C for 30 s, extension at 72 °C for 40 s, and a final extension cycle at 72 °C for 10 min. The PCR products were sequenced using the Sanger method (Macrogen Inc., Seoul, Republic of Korea), and the sequences were compared with those obtained by HTS.

### 2.5. Confirmation of Two New Viroid Species

The two genome drafts of each viroid (MOVd and OVd-2), assembled by HTS, were used to design primers that would allow the amplification of the entire genome by RT-PCR in two different reactions for each of them. For this purpose, forward and reverse primers (Table 1) were designed from the 5′ end to the 5′ end in opposite directions of the genome at two different positions to generate overlapping amplicons covering the entire genome of each viroid [9].

Reverse transcription was performed as previously described, using a portion of the total RNA that underwent HTS as the template. One microliter of cDNA was added to the PCR mixture as previously indicated. The amplification conditions for the four pairs of primers were as follows: initial incubation at 95 °C for 3 min, followed by 35 cycles of denaturation at 95 °C for 30 s, annealing at 55 °C for 30 s, and extension at 72 °C for 30 s; and a final extension cycle at 72 °C for 10 min. The PCR products were purified and bidirectionally sequenced using the Sanger method (Macrogen Inc., Seoul, Republic of Korea).

### 2.6. RT-PCR Detection of Viruses and Viroids

From June to December 2023, a targeted sampling was conducted in the eastern region of the State of Mexico, one of the main cactus pear-producing areas in the country (Table 2). Seventy-five cladodes showing putative viral symptoms were collected (Figure 1): nopalitos (*n* = 36), cactus pears (*n* = 35), and xoconostles (*n* = 4). Seven asymptomatic nopalitos samples, two cactus pear samples, and two wild cactus pear samples were collected. Total RNA extraction was performed using the SV Total RNA Isolation System kit (Promega, Madison, WI, USA), and RNA quality and quantity were verified using a NanoDrop spectrophotometer (Thermo Fisher Scientific, Waltham, MA, USA). RT-PCR was carried out as previously described using specific primers for the detection of the three viruses and three viroids (Table 1).

## 3. Results

### 3.1. HTS Data

The libraries from the two selected composite samples of nopalitos and cactus pears for fruit production yielded 24,089,241 and 24,123,293 paired reads of 150 bp, respectively. The reads were clean and adapter-free and were subjected to quality control by a sequencing service provider [32]. These RNA reads were deposited in the NCBI Sequence Read Archive and assigned the accession numbers SRX24164751 and SRX24164750 (BioProject Number PRJNA1096702). Metagenomic analysis revealed different viruses and viroids in the transcriptomes of both cactus pears, which are described below.

#### Known Viruses and Viroids That Infect *Opuntia*

The presence of three previously reported viruses and one viroid was detected and confirmed [11,12]. The identified viruses were OV2, CCV-1, and OPV-A. Following genome reconstruction, they were annotated for their coding regions and deposited in GenBank (Table 3). The detected viroid was OVd-1, and its genome was also deposited in GenBank (Table 3). In the nopalitos sample, OV2, CCV-1, OPV-A, and OVd-1 were detected, while in the cactus pear for fruit production sample, OV2, CCV-1, and OPV-A were detected. The presence of these viruses and viroids was confirmed through RT-PCR and Sanger sequencing.

### 3.2. Phylogenetic Analysis and Genetic Diversity of the Detected Viruses

#### 3.2.1. Opuntia Virus 2 (Genus *Tobamovirus*)

In the two samples subjected to HTS, the complete genomes of OV2 and both isolates [EM_T2 (PP579659) and EM_T1 (PP579656)] had the highest nucleotide identity (98.98%) with a previously obtained isolate from nopalitos (NC_040685.2) in Mexico [11]. 

The average nucleotide coverage of the OV2 genome was 12,492x for the EM_T2 isolate and 1712x for the EM_T1 isolate (Appendix A).

Phylogenetic analysis grouped the two OV2 isolates obtained in this study into a clade with other genomes of this virus that are phylogenetically related to tobamoviruses that naturally infect cacti (Figure 2).

#### 3.2.2. Cactus Carlavirus 1 (Genus *Carlavirus*)

In the samples of cactus pears for fruit production and nopalitos, CCV−1 was detected (designated as isolates EM_C1 and EM_C2, respectively). The average nucleotide coverage over the genome of the EM_C2 isolate was 187,006x, and that of the EM_C1 isolate was 3.52x (Appendix A). Considering a 95% nucleotide identity as a tentative cutoff to determine if it is a divergent isolate, the nopalitos isolate [EM_C2 (PP579657)] is considered a divergent isolate with a 92.14% nucleotide identity compared to the closest available genome in GenBank (Table 3). 

Phylogenetic analysis grouped this isolate into a clade with other CCV-1 isolates (Figure 3). The EM_C1 isolate, obtained from cactus pear, lacked sufficient sequencing reads for complete genome reconstruction (Appendix A) and was therefore excluded from the phylogenetic analysis.

#### 3.2.3. Opuntia Potexvirus A (Genus *Potexvirus*)

OPV-A was detected in both cactus pears for fruit production and nopalitos (designated as isolates EM_A1 and EM_A2, respectively). The nucleotide identity percentages were 85.50% for EM_A2 and 90.1% for EM_A1 compared to the single reference genome available in GenBank (OQ240443.1). Based on the nucleotide identity percentage, both OPV-A isolates are considered highly divergent new isolates.

The average nucleotide coverage of the EM_A2 isolate over the reference genome was 169,648x and 33,437x for the EM_A1 isolate (Appendix A).

Phylogenetic analysis grouped the two isolates from this study into a clade with the only OPV-A sequence available in GenBank (Figure 4).

#### 3.2.4. Opuntia Viroid 1 (Putative Genus *Apscaviroid*)

Opuntia viroid 1 was detected only in nopalitos (designated as isolate EDMEX-V1) (Table 3), showing a 98.54% nucleotide identity with the single reference genome in GenBank and a coverage of 99x (Appendix A).

### 3.3. Description of Two New Viroid Species

In addition to the mentioned viruses and viroid, two new viroid species were also detected (Table 4), one belonging to the genus *Pospiviroid* and the other to the genus *Apscaviroid*, based on the species demarcation criteria established for each genus [33].

#### 3.3.1. Opuntia Viroid 2 (Genus *Apscaviroid*)

Opuntia viroid 2 (OVd-2) was detected in the cactus pear fruit production sample, with an average nucleotide coverage over the genome of 831x (Appendix A). The primary structure was obtained by direct sequencing of overlapping amplicons obtained by RT-PCR with primers 90F/89R and 208F/207R (Figure 5B, Table 1), which allowed determination of the circular nature of the RNA (Figure 5B). The consensus sequence was deposited in GenBank with the accession number PP579661.1. The OVd−2 genome consists of 319 nucleotides with a G + C content of 66.6%. 

The secondary structure consists of a rod-shaped molecule containing a conserved central region (CCR) (Figure 5A), a key criterion for classifying a new viroid as a member of the family *Pospiviroidae* [23]. Considering the type of CCR and the presence of a conserved terminal region (TCR) (Figure 5A), OVd-2 is classified as a species of the genus *Apscaviroid.*

In this study, autonomous replication of OVd-2 was not demonstrated; however, the molecular characteristics of the viroid RNAs reported strongly support that it is a new viroid species. A phylogenetic analysis was conducted, which included the representative sequence of OVd-2 (EDMEX-S1, GenBank accession number PP579661) and all available reference sequences (RefSeq) in GenBank for current members belonging to the family *Pospiviroidae* (including an OVd-1 sequence) and the sequence of the Peach latent mosaic viroid (family *Avsunviroidae*) as an outgroup. OVd-2 clustered with current members of the genus *Apscaviroid* (Figure 6). Pairwise comparison-based SDT analysis also shows that none of the viroid sequences available in GenBank were more than 80% similar across the entire genome (Figure 7).

#### 3.3.2. Mexican Opuntia Viroid (Genus *Pospiviroid*)

The Mexican opuntia viroid (MOVd) was found only in the nopalitos sample. Its genome consists of 407 nucleotides with a G + C content of 68.1%. The average nucleotide coverage across the genome was 253x (Appendix A), and by directly sequencing the overlapping amplicons obtained by RT-PCR with the primer pairs 87F/86R and 307F/306R (Table 1, Figure 8A), the circular nature of this viroid RNA was determined (Figure 8B). The consensus sequence has been deposited in GenBank with the accession number PP579660.

The molecular structures of viroid RNAs previously reported and the results of the pairwise comparison-based SDT analysis, in which none of the complete genome sequences of viroids available in GenBank had more than 90% similarity (Figure 7), demonstrated that MOVd is a new viroid. Additionally, the phylogenetic analysis (Figure 6) confirmed that MOVd should be considered a new species of the genus *Pospiviroid.*

### 3.4. Relative Abundance of Viruses and Viroids Detected by HTS in the Eastern Cactus Pear-Producing Area of the State of Mexico

Eighty-six samples of cladodes with different symptoms associated with viruses collected in the municipalities of Otumba, Axapusco, Temascalapa, Nopaltepec, and Teotihuacan were analyzed for the detection of OV2, CCV-1, OPVA, OVd-1, OVd-2, and MOVd. OV2 was detected in 91% of the samples (37 from cactus pears for fruit production, 39 from nopalitos, 2 xoconostles, and 1 wild cactus pear), followed by CCV-1 in 66% of the samples (8 from cactus pears, 43 from nopalitos, and 2 wild cactus pears), and finally OPVA in 50% of the positive samples (22 from cactus pears, 12 from nopalitos, 2 xoconostles, and 1 wild cactus pear). For viroids, OVd-2 was detected in 12 cactus pear samples, OVd-1 in 5 nopalitos samples and 1 cactus pear sample, and MOVd was detected in 2 nopalitos samples. 

All samples showing symptoms were positive for at least one virus/viroid, except for one xoconostle sample, which tested negative for the viruses and viroids analyzed. The most frequent mixed infections in symptomatic samples were OV2 + CCV-1 (27 samples) and OV2 + OPV-A (10 samples). On the other hand, in the asymptomatic samples, all samples were positive for at least one virus; mixed infection was detected in only three samples of nopalitos (OV2 + CCV-1) and two samples of wild cactus pears (OV2 + CCV-1; CCV-1 + OPV-A) (Appendix A). 

## 4. Discussion

In this study, we report on the viruses and viroids detected in cactus pears, cultivated for both fruit production and the harvest of nopalitos, using HTS in the eastern region of the State of Mexico, Mexico. Two of these viroids are new species, one belonging to the *Posviroid* genus and the other to the *Apscaviroid* genus. In 2023, it was reported that OVd-1 (from the *Apscaviroid* genus) infects nopalitos in Mexico City, making it the first viroid reported in cacti worldwide [12]. One of the criteria used to demarcate new viroid species is having a sequence identity of less than 90%. Therefore, OVd-2 should be considered a new species within the *Apscaviroid* genus, and MOVd should be considered a new species of the *Pospiviroid* genus.

Additionally, the conserved structural domains of MOVd are very similar to those of the reference genomes of the potato spindle tuber viroid (PSTVd) and Iresine viroid 1 (IrVd-1) (*Pospiviroid* genus); the CCR of MOVd is identical to that of IrVd-1, while there are slight modifications in the TCR and CCR compared to those of PSTVd. In the case of OVd-2, the conserved structural domains were identical to those in the reference genome of the apple scar skin viroid (ASSVd) (*Apscaviroid* genus) (Table 5).

Due to the presence of mixed infections of viruses and viroids in 82% of the samples analyzed individually by RT-PCR (Appendix A), it was not possible to exclusively associate the symptoms with either of them. This finding suggests a strong correlation between the presence of symptoms and infection by viruses or viroids. Interestingly, the most frequent mixed infections in symptomatic samples were OV2 + CCV-1 (27 samples) and OV2 + OPV-A (10 samples). This indicates that these viruses have a high joint prevalence in symptomatic plants, which could be indicative of synergistic interactions that enhance symptom expression. On the other hand, in the asymptomatic samples, all tested positive for at least one virus. However, the frequency of mixed infections was considerably lower than that in symptomatic samples. This contrast suggests that the presence of a single virus may not be sufficient to cause visible symptoms; coinfection with multiple viruses could be decisive in symptom manifestation. There is limited understanding of the role of virus/viroid coexistence, warranting a study on this interaction [34]. In Mexico, at least three cases of mixed viroid infections have been reported: the Mexican papita viroid (MPVd)/tomato chlorotic dwarf viroid and MPVd/tomato severe leaf curl virus in tomatoes under greenhouse conditions and the hop stunt viroid/citrus exocortis viroid and citrus tristeza virus in oranges [34]. The symptoms of plants with mixed infections were more severe than those of plants with individual infections in all cases [34]. Therefore, experiments on pathogenicity need to be performed, along with studies on the distribution, transmission modes, and effects on the production and quality of nopalitos or fruit production caused by these new viroids and viruses in cactus pears. 

The detection of these new viroids in cactus pear cultivation reinforces the hypothesis that Mexico, being the center of origin and domestication of *O. ficus-indica* (the most cultivated cactus species in the world) and other species of *Opuntia* [1], is a geographical region of viroid origin and possesses considerable biodiversity including endemic species that affect important crops such as tomatoes and other commercial products such as avocados [35,36]. The viroids detected in this study are among the most common viroids, such as the tomato planta macho viroid, Mexican papita viroid, and avocado sunblotch viroid, which are considered endemic to Mexico [34].

Moreover, of the detected viruses, CCV-1 and OPV-A were the most divergent (Figure 3 and Figure 4), while the OV2 isolates consistently clustered with previously reported isolates, forming a clade with viruses that naturally infect cacti (Figure 6).

It is known that tobamoviruses, such as OV2, are easily transmitted mechanically [37]; so, this is an important aspect to consider during crop management since many cultural practices in both cactus pears and nopalitos involve the use of tools for pruning and harvesting that are not disinfected among plants.

OPV-A was previously detected in nopalitos in Mexico City, but its range of hosts and economic impact on cactus pear production are still unknown.

CCV-1 was detected alongside cactus carlavirus 2 (CCV-2) for the first time in the United States, asymptomatically, in a plant of the genus *Epiphyllum* (hybrid ‘Professor Ebert’) [38]. In Mexico, CCV-1 was detected in 93 out of 129 nopalitos samples collected in the states of Morelos, Mexico, Hidalgo, and Mexico City [12]. On the other hand, it is known that carlaviruses predominantly infect herbaceous plants; many cause latent or asymptomatic infections and are transmitted by aphids in a nonpersistent manner and by whiteflies [38]. However, none of these insect groups are significant pests of cactus pears, hence transmission is not yet associated with any vector insect.

The vegetative and mechanical transmission of CCV-1, OV2 and OPV-A can contribute to their distribution in new cactus pear plantations not only in the study area but also throughout the country as the State of Mexico is a provider of propagative cactus pear material. Therefore, phytosanitary measures should be implemented to help reduce the risk of the spread and movement of infected material carrying viruses or viroids.

Previously, a study was conducted to understand the virome of nopalitos in the central region of Mexico, which included the states of Morelos, Hidalgo, Mexico City, and the State of Mexico. However, the two new viroid species reported in the present study were not detected in the State of Mexico [12]. This suggests the need to consider more sampling sites and process a greater number of samples to generate more robust information. Additionally, in this study, we report for the first time the presence of OPV-A and OV2 in xoconostles and for the first time the presence of CCV-1 and OPV-A in cactus pears.

Subsequent research on OVd-1, OVd-2, and MOVd should focus on understanding their host range, environmental conditions, or crop management practices that favor asymptomatic conditions, symptoms associated with individual and mixed infections, among other aspects. Although the main characteristic of viroids is their autonomous replication, their ability to be transmitted through grafting alone is not sufficient to demonstrate this feature [9]; therefore, mechanical or biolistic transmission tests are necessary. Additionally, further investigations into other aspects of their transmission, such as the potential presence of insect vectors, will be needed to deepen our understanding of these viroids. 

In this study, we confirmed the potential of HTS to provide an ideal methodology for determining the complete infection status of a plant by viruses and viroids. This technology does not depend on the availability of genomic sequence information such as RT-PCR, and it is limited only by the integrity of the reference database against which the sequences are compared [9] as well as the correct use of different programs for bioinformatic analysis.

## 5. Conclusions

Using HTS, the presence of a new viroid of the *Pospiviroid* genus and a second *Apscaviroid* in cactus pears is reported for the first time. Additionally, Opuntia virus 2 was the most abundant virus in the study area. Finally, mixed infection was detected in 82% of the samples analyzed. To prevent the potential spread of these viruses and viroids in new cactus pear plantations, a program for producing virus- and viroid-free mother plants should be implemented to ensure the health of the propagative material.

## Figures and Tables

**Figure 1 viruses-16-01177-f001:**
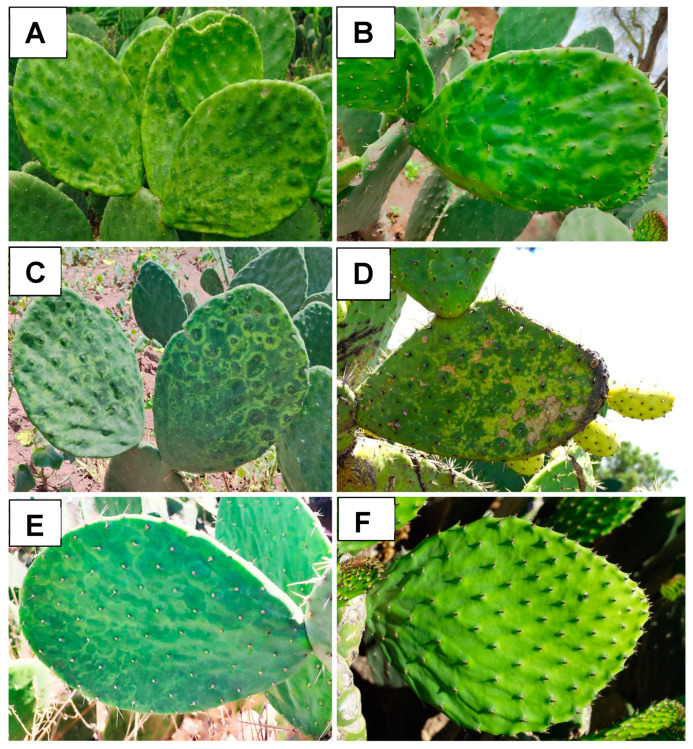
Symptoms associated with viruses in different cactus pears. (**A**,**B**), Chlorosis with green islands in nopalitos; (**C**), Irregular chlorotic patterns around the spines in nopalitos; (**D**), Mottling and chlorotic spots in cactus pears for fruit production. (**E**), Ring spot symptoms in xoconostles; (**F**) Symptomless nopalitos.

**Figure 2 viruses-16-01177-f002:**
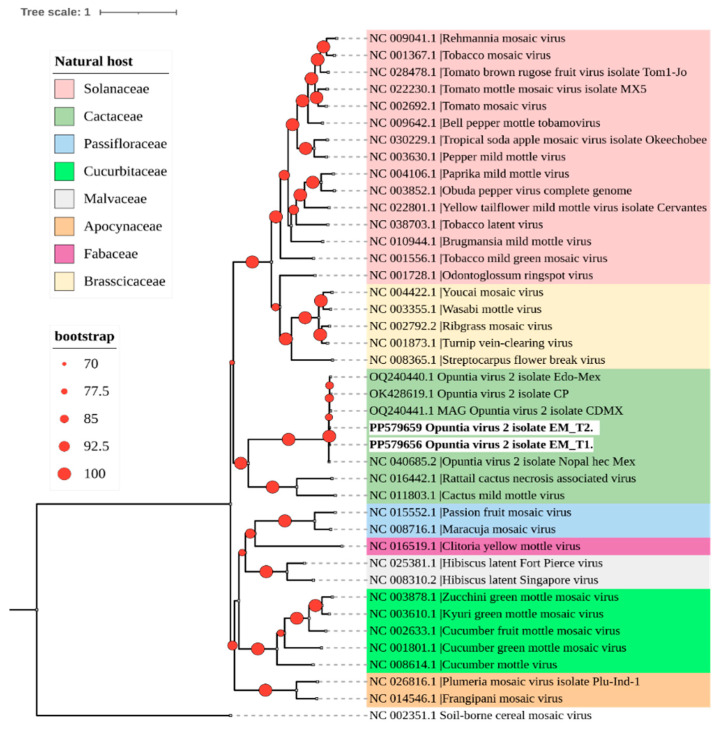
A phylogenetic tree was constructed using the maximum likelihood method with complete genome sequences of the *Tobamovirus* genus obtained from GenBank (accession numbers shown). The tree was built with IQtree 2.3.1 using a color code to represent the different botanical families that are natural hosts of each virus. The identified OV2 sequences in this study are highlighted in bold. The sequence of soil-borne cereal mosaic virus (genus *Furovirus*) was used as an outgroup. Circles on branches indicate UFBoot support values >70%.

**Figure 3 viruses-16-01177-f003:**
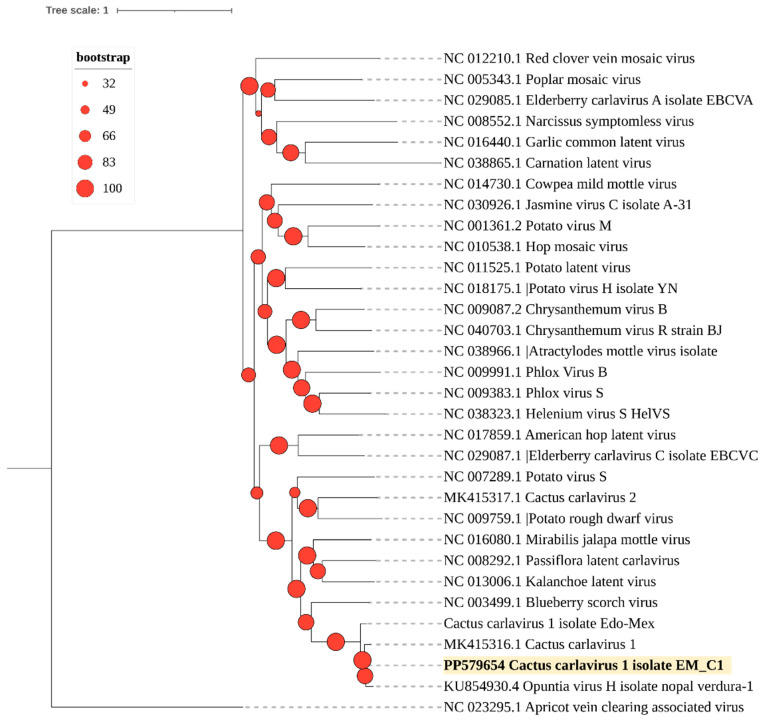
A phylogenetic tree was constructed using the maximum likelihood method with complete genome sequences of the *Carlavirus* genus obtained from GenBank (accession numbers shown). The tree was built with IQtree 2.3.1. The CCV-1 sequence identified in this study is highlighted in bold. The sequence of the apricot vein clearing-associated virus (genus *Prunevirus*) was used as an outgroup. Circles on branches indicate UFBoot support values >70%.

**Figure 4 viruses-16-01177-f004:**
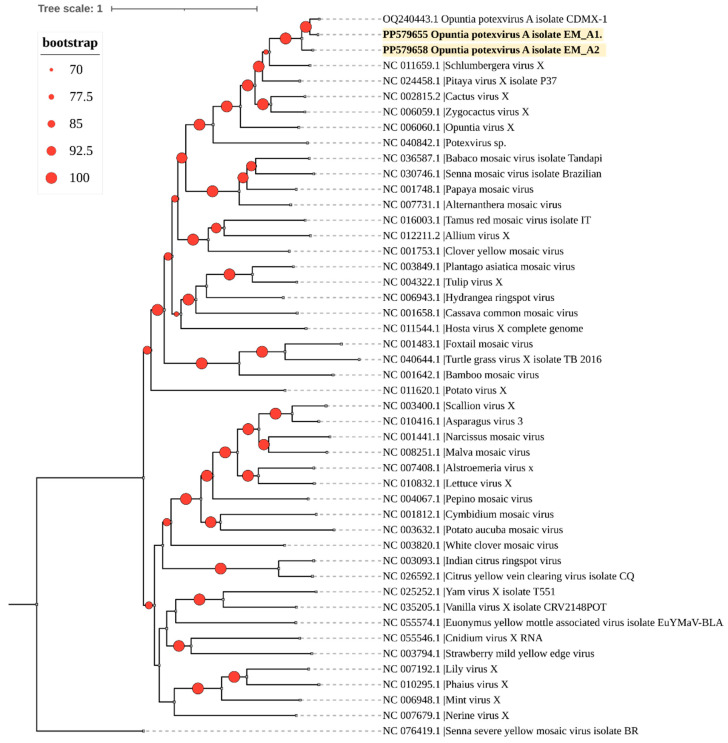
A phylogenetic tree was constructed using the maximum likelihood method with complete genome sequences of the *Potexvirus* genus obtained from GenBank (accession numbers shown). The tree was built with IQtree 2.3.1. The identified OPV-A sequences in this study are highlighted in bold. The sequence of the Senna severe yellow mosaic virus (genus *Allexivirus*) was used as an outgroup. Circles on branches indicate UFBoot support values >70%.

**Figure 5 viruses-16-01177-f005:**
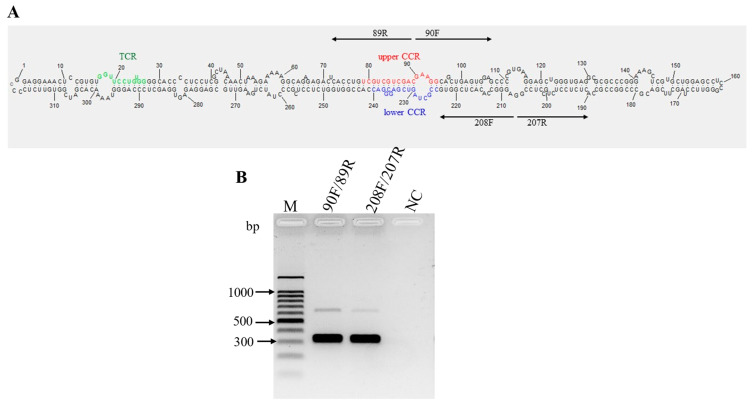
Characterization of the predicted secondary structure and conserved structural elements of Opuntia viroid 2 (OVd-2). (**A**), Nucleotides in the upper and lower conserved central region (CCR) are indicated in red and blue, respectively. The nucleotides in green indicate the terminal conserved region (TCR) structural element. The two pairs of arrows pointing in opposite directions represent the primers used for amplifying the complete genome of the viroid; (**B**), The circular RNA structure of OVd-2 was determined using RT-PCR with the primer sets F90/89R and 208F/207R, each adjacent to the other in opposite directions (‘F’ = forward primers; ‘R’ = reverse primers). A healthy nopalitos sample was used as a negative control (NC). M, 100 bp DNA ladder (Promega, Madison, WI, USA).

**Figure 6 viruses-16-01177-f006:**
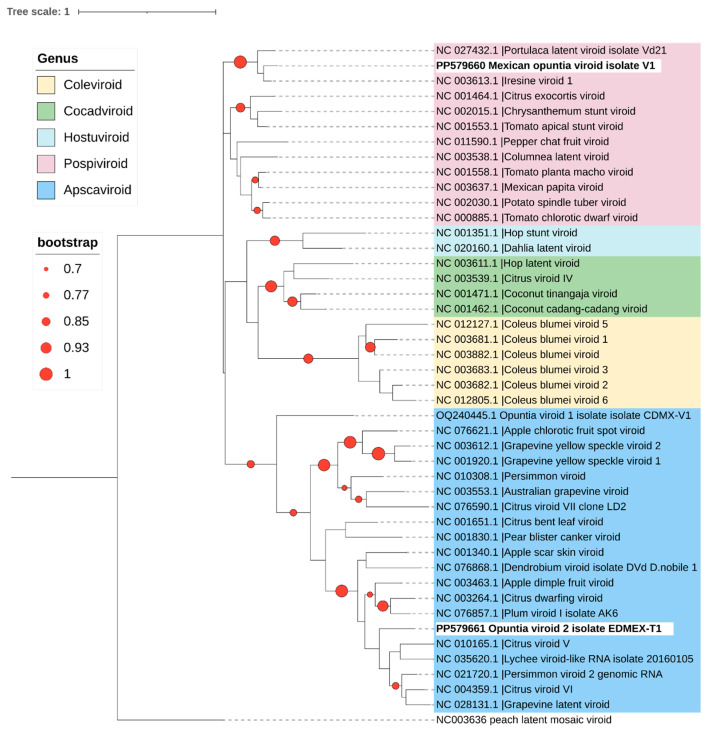
Phylogenetic relationships of Opuntia viroid 2 (OPVd-2), Mexican opuntia viroid (MOVd) (highlighted in bold), and all current RefSeq viroid sequences available in GenBank for the family *Pospiviroidae.* The phylogenetic tree was built with 1000 bootstrap replicates using maximum likelihood method and the Jukes–Cantor model in MEGA X [9,23]. A color code was used to represent the different genera within this family. Peach latent mosaic viroid (PLMVd) (family *Avsunviroidae*) was included as an outgroup. Circles on branches indicate bootstrap values >70% (generated from 1000 replicates).

**Figure 7 viruses-16-01177-f007:**
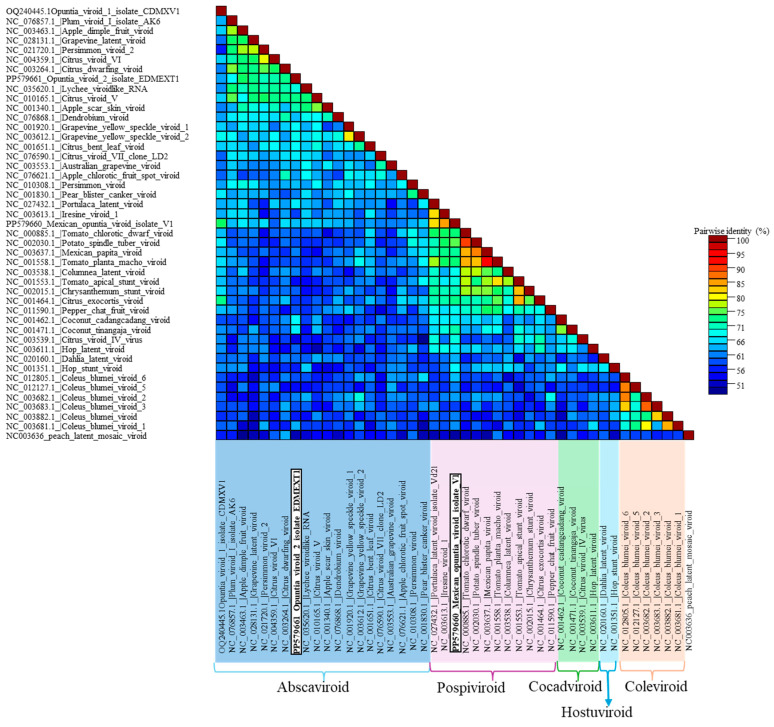
The pairwise identity frequency distributions obtained with the Sequence Demarcation Tool (SDT) [25] indicated that Opuntia viroid 2 (OVd-2, sequence PP579661) and the Mexican opuntia viroid (MOVd, sequence PP579660) are new viroid species. The sequences of OVd-2 and MOVd had less than 80% and 90% identity, respectively, with any other known viroid sequence across the entire genome. Different genera within the family *Pospiviroidae* are indicated. Peach latent mosaic viroid (PLMVd, NC_003636) (family *Avsunviroidae*) was included as an outgroup.

**Figure 8 viruses-16-01177-f008:**
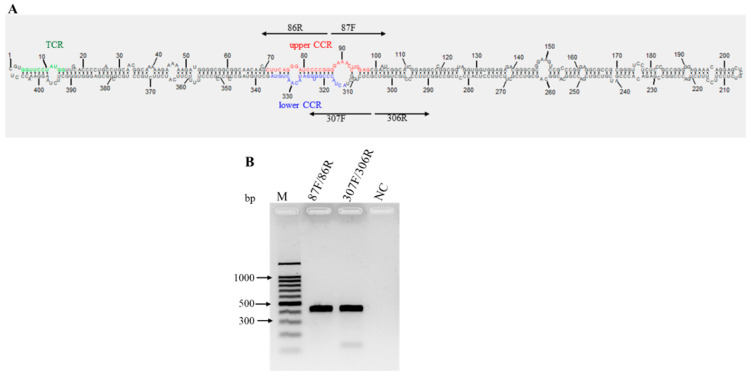
Characterization of the predicted secondary structure and conserved structural elements of Mexican opuntia viroid (MOVd). (**A**), Nucleotides in the upper and lower conserved central region (CCR) are indicated in red and blue, respectively. The green nucleotides indicate the terminal conserved region (TCR) structural elements. The two pairs of arrows pointing in opposite directions represent the primers used for full genome amplification; (**B**) The circular nature of MOVd RNA was confirmed through RT-PCR with the primer sets F87/86R and 307F/306R, each adjacent to the other in opposite directions (‘F’ = forward primers; ‘R’ = reverse primers). A healthy nopalitos sample was used as a negative control (NC). M, 100 bp DNA ladder (Promega, Madison, WI, USA).

**Table 1 viruses-16-01177-t001:** Primers used for the validation and detection of viruses/viroids in cactus pears (for fruit production and nopalitos) from the eastern region of the State of Mexico.

Virus/Viroid	Primer	Sequence (5′-3′)	Amplicon (bp)	Region
Opuntia potexvirus A	OPVA-RepF	AAGCTCGCAGCATCCATCAA	482	Viral replicase
Opuntia potexvirus A	OPVA-RepR	GGGTGAAGGGACGGTAGTTG		
Cactus carlavirus 1	CCV-1F	AATGGGCGCCTTTAGGTTCA	559	Capsid protein
Cactus carlavirus 1	CCV-1R	AATTCCAAGCTCCCGTCAGG		
Opuntia virus 2	OV2-F	CTTCCAAGAGTTCTAGCGCCT	609	Capsid protein
Opuntia virus 2	OV2-R	ACCTGCAGGATTACCACCAC		
Opuntia viroid 1	OPVd_IF	GACGGAGCGTCGAGAAGTAG	412	Complete genome
Opuntia viroid 1	OPVd_IR	GCC GGC GCC GAA GCC CGA G		
Mexican opuntia viroid	307F	TCTGGCTACTACCCGGTGG	407	
Mexican opuntia viroid	306R	GCGACCAGCAGGGGAAG		Complete genome
Mexican opuntia viroid	86R	CCGGGGATCCCTGAAG	407	
Mexican opuntia viroid	87F	GGAAACCTGGAGCGAACTC		Complete genome
Opuntia viroid 2	F90	GAAGGCAGCTGAGTGGAG	319	
Opuntia viroid 2	R89	GTCGACGACGACAGGTGA		Complete genome
Opuntia viroid 2	208F	AGGGCCACACTCGGTG	319	Complete genome
Opuntia viroid 2	207R	CCGGAGGCAGAGGAGAG		

**Table 2 viruses-16-01177-t002:** Locations sampled for the detection of viruses and viroids in different types of cactus pear in the eastern region of the State of Mexico.

Municipality	Location	Type of Cactus Pear			
		Fruit	Nopalitos	Xoconostles	Wild Cactus Pears
Axapusco	Cuautlacingo	3	38	0	0
	Axapusco	4	0	0	0
Temascalapa	Santa Ana Tlachiahualpa	11	1	0	0
Nopaltepec	San Felipe Teotitlán	19	0	4	2
Teotihuacán	Teotihuacán	0	4	0	0
Total		37	43	4	2

**Table 3 viruses-16-01177-t003:** Comparison of total or partial genome sequences of the viruses and viroids detected through HTS in different cactus pear samples with the most similar reference sequence available in NCBI’s GenBank.

HTS Sample	Virus/Viroid Detected *	GenBank Accession Number	Isolation	Genome Segment	Reference Sequence Accession Number (NCBI GenBank)	% Identity with Reference
Nopalitos	CCV-1	PP579657	EM_C2	complete	KU854930.4	92.14%
	OPV- A	PP579658	EM_A2	complete	OQ240443.1	85.50%
	OV2	PP579659	EM_T2	complete	NC_040685.2	98.98%
	OVd-1	PP579662	EDMEX-V1	complete	OQ240445.1	98.54%
Cactus pears for fruit production	CCV-1	PP579654	EM_C1.	** RdRp	KU854930.4	88.83%
	OPV-A	PP579655	EM_A1	complete	OQ240443.1	90.16%
	OV2	PP579656	EM_T1	complete	NC_040685.2	98.98%

* CCV-1: Cactus carlavirus 1; OPV-A: Opuntia potexvirus A; OV2: Opuntia virus 2; OVd-1: Opuntia viroid 1. ** RNA-dependent RNA polymerase

**Table 4 viruses-16-01177-t004:** New viroid species detected by HTS in cactus pears for fruit production and nopalitos.

HTS Sample	Viroid	GenBank Accession Number	Isolate	Closest Viroid in BLASTn Analysis	Reference Sequence, Accession Number (NCBI GenBank)	% Identity with Reference (% Query Cover)
Nopalitos	Mexican opuntia viroid	PP579660	V1	Iresine viroid 1	OM108483.1	83.33 (99)
Cactus pears for fruit production	Opuntia viroid 2	PP579661	EDMEX-T1	Grapevine latent viroid	MG770884.1	82.81 (38)

**Table 5 viruses-16-01177-t005:** Location of conserved domains of viroids in the reference sequence of the *Pospiviroid* and *Apscaviroid* genera compared to the new viroid species detected in this study. Discrepancies in the sequences are underlined and in bold.

Viroid *	TCR	CCR Upper Strand	CCR Lower Strand
MOVd (PP579660.1)	GGUUCC**UG**UGG	CUUCAGGGAUCCCCGGGGAAACCUGGAG	ACUACCCGGUGGA**U**ACAACUG**U**AGCU
PSTVd (NC_002030.1)	GGUUCCUAUGG	CUUCAGGGAUCCCCGGGGAAACCUGGAG	ACUACCCGGUGGAAACAACUGAAGCU
IrVd−1 (NC_003613.1)	GGUUCCAAUGG	CUUCAGGGAUCCCCGGGGAAACCUGGAG	ACUACCCGGUGGAUACAACUGUAGCU
OVd−2 (PP579661.1)	GGUUCCUGUGG	UCGUCGUCGACGAAGG	CCGCUAGUCGAGCGGAC
ASSVd (NC_001340)	GGUUCCUGUGG	UCGUCGUCGACGAAGG	CCGCUAGUCGAGCGGAC

* MOVd: Mexican opuntia viroid; PSTVd: potato spindle tuber viroid; IrVd-1: Irasine viroid 1; OVd-2: Opuntia viroid-2; ASSVd: apple scar skin viroid.

## Data Availability

The genomic sequences of the viruses and viroids described in this study are available in the GenBank database under the accession numbers PP579654-PP579661 and BioProject No. PRJNA1096702.

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
