# Peer review of "High-Throughput Sequencing Reveals New Viroid Species in Opuntia in Mexico"

_viruses, 2024, doi:10.3390/v16081177_

Round 1

Reviewer 1 Report

Comments and Suggestions for Authors

The paper of Ortega et al. (2024) presents a viromic study on prickly pear cactus (Opuntia ficus-indica) from two composite samples collected in the State of Mexico and presenting virus-like symptoms. They detected three known viruses and one known viroid, and they also identified two novel viroid species. Phylogenetic analyses were performed on the viruses and viroids, and genomic structure was determined for the novel viroids. Then, a field survey was carried out on 86 nopal samples (cultivated and wild) in order to assess the virus and viroid prevalence by RT-PCR. This study was well designed, combining untargeted HTS and targeted RT-PCR detection in order to better understand the virome associated with nopal in Mexico, and it could extend the list of viroids infecting these plants.

General comments:

The study is well presented, clear, concise and relevant for the field of plant viruses infecting cacti. It describes with details the different stages of a virus metagenomic study, and the confirmation of the results by RT-PCR and Sanger sequencing. Some details would be needed about the type of virus-like symptoms observed in the composite samples. Are they the same symptoms as described in Figure 1 for the field survey ? Moreover, adding some controls in the HTS libraries would have been helpful to deal with viruses identified with a low number of reads (e.g. EM_C1), but RT-PCR confirmation could compensate it.

The figures and the tables are clear and they bring more information to virologists, in particular for the genomic structure of the novel viroid species identified allowing their classification, and for the phylogeny of the different virus and viroid taxa detected in the HTS data. The references used are recent, relevant and appropriate for the field of plant virology and metagenomics.

In addition to the HTS study which identified a series of known and novel viruses, a field survey was performed to analyze the virus prevalence and co-infection in wild and cultivated nopal samples (symptomatic or not). This survey is well described and presented but should be detailed a bit more, in both results and discussion sections. For instance, more details about the mixed-infection (proportion, main combination of mixed-infection observed), or about any difference in virus prevalence between wild and cultivated nopal, symptomatic or non-symptomatic,… There are a lot of interesting information in the Supplementary Table 1 but they could be interpreted more.

Besides these few points to address, this is globally a very good study worth publishing and will improve our knowledge on cactus viruses and viroids.

Specific comments:

L83: Provide more information about virus-like symptoms

L157-158 (Table 1): I cannot see the primers for Mexican opuntia viroid. Is it the Opuntia viroid III ? If it is, please change the name into Mexican opuntia viroid in this table.

L199: Typo “subjected” (c missing).

L216-217 (Table 3): Please complete the legend for OV2: Opuntia virus 2

L376-386 : Provide a few lines about the virus prevalence and co-infection in symptomatic vs. non-symptomatic samples, wild vs. cultivated prickly pear cactus.

L396-400 : The reference to Table 5 seems to be missing in main text.

L408-418: Please complete the discussion about the prevalence and co-infection (Cf. L376-386).

Comments on the Quality of English Language

The quality of English is very good, some typos were found but the study can totally be understood. Please just double-check the manuscript in case of any other typos.

Author Response

Thank you very much for taking the time to review this manuscript. Please find the detailed responses below and the corresponding revisions/corrections highlighted in red in the re-submitted files. 

To make the manuscript easier to understand, we updated the names to "cactus pear for fruit production" to refer to nopal cultivated for fruit production (tunas) and "nopalitos" to refer to nopal cultivated as a vegetable. This will help the reader better understand the two types of cactus grown in Mexico.

The suggestions from the general comments were addressed in the new version of the manuscript.

Specific comments are described below.

Specific comments:

Comments 1: L83: Provide more information about virus-like symptoms

Response 1: This suggestion was added

Comments 2. L157-158 (Table 1): I cannot see the primers for Mexican opuntia viroid. Is it the Opuntia viroid III ? If it is, please change the name into Mexican opuntia viroid in this table.

Response 2: We have updated the table 1 and this error has been corrected.

Comments 3. L199: Typo “subjected” (c missing).

Response 3: This suggestion was added

Comments 4. L216-217 (Table 3): Please complete the legend for

Response 4: This suggestion was added

Comments 5: L376-386: Provide a few lines about the virus prevalence and co-infection in symptomatic vs. non-symptomatic samples, wild vs. cultivated prickly pear cactus.

Response 5: This suggestion was added

Comments 6: L396-400: The reference to Table 5 seems to be missing in main text.

Response 6: The reference in Table 5 has already been added.

Comments 7: L408-418: Please complete the discussion about the prevalence and co-infection (Cf. L376-386).

Response 7: We have added the discussion as requested.

Comments 8: Comments on the Quality of English Language

The quality of English is very good, some typos were found but the study can totally be understood. Please just double-check the manuscript in case of any other typos

Response 8: The English was reviewed by a native speaker. The changes made are in red throughout the document

Reviewer 2 Report

Comments and Suggestions for Authors

The revised version of the paper includes most of the comments made by the reviewers. Although some aspects that were not taken into consideration are important, the authors' explanations allow for the publication of the paper in its current form.

Author Response

Thank you very much for taking the time to review this manuscript.